# Metabolic, Hematological, and Functional Health in Adults with Down Syndrome and Significance of Parental Health Literacy: A Cross-Sectional Study

**DOI:** 10.3390/healthcare13101212

**Published:** 2025-05-21

**Authors:** Petra Rajkovic Vuletic, Marijana Geets-Kesic, Anamarija Jurcev-Savicevic, Nurjanah Nurjanah, Barbara Gilic

**Affiliations:** 1Faculty of Kinesiology, University of Split, 21000 Split, Croatia; petra.rajkovic@kifst.eu (P.R.V.); markes@kifst.hr (M.G.-K.); barbara.gilic@kifst.eu (B.G.); 2Faculty of Kinesiology, University of Zagreb, 10000 Zagreb, Croatia; 3Teaching Institute of Public Health of Split Dalmatian County, 21000 Split, Croatia; 4University Department of Health Studies, University of Split, 21000 Split, Croatia; 5School of Medicine, University of Split, 21000 Split, Croatia; 6Faculty of Health Sciences, Universitas Dian Nuswantoro, Semarang 50131, Indonesia; nurjanah@dsn.dinus.ac.id

**Keywords:** disability, trisomy, physical activity, well-being, blood parameters, therapeutic exercise, recreation

## Abstract

**Background/Objectives**: The evaluation of metabolic and physiological health indicators in people with Down syndrome (DS) is crucial, since these people are more prone to metabolic problems. However, there is limited scientific evidence regarding the health status and health literacy (HL) of adults with DS and their legal guardians. This study aimed to assess the health status of adults with DS and determine the HL levels of their legal guardians. **Methods:** Eighteen adults (11 females, 7 males) with DS aged 28.64 ± 9.01 years were tested for health status, and their legal guardians completed the HL survey. Gender differences in all study variables were checked by *t*-tests for independent samples and Cohen’s D effect sizes (ESs). Differences in all study variables between parents with low and adequate HL were calculated via receiver operating characteristic curves. **Results:** Males were overweight, whereas females were obese (mean BMI = 26.51 and 30.10 for males and females, respectively). Females had higher high-density lipoprotein concentrations (large ES), whereas males had higher hematocrit and hemoglobin concentrations (large ES). Hematological parameters were the most significant variables that differed between parents with limited and adequate HL status (AUC = 0.79–0.87). **Conclusions:** These findings suggest that in the absence of severe comorbidities, adults with DS may achieve stable health profiles, particularly when supported by structured physical activity and informed caregiving. The influence of parental HL on health parameters points to the potential for parent-targeted health education to improve health outcomes and promote autonomy in individuals with DS through supported decision-making. Thus, our findings highlight the need for greater investment in caregiver and parental health education and systemic support to optimize health outcomes in adults with DS. Future research should explore interventions aimed at improving parental HL and examine the extent to which these efforts translate into improved health outcomes for people with DS.

## 1. Introduction

Down syndrome (DS) (Q90) is a disorder of chromosomes (i.e., the presence of either the entire or part of an additional duplication of chromosome 21) and is the most prevalent chromosomal disorder in humans [1]. Indeed, between 2011 and 2015, there were an estimated 8031 live births of children with DS in Europe [2]. DS is characterized by various health deficits, including intellectual disability and mental health problems, such as autism spectrum disorder and attention-deficit/hyperactivity disorder; physical abnormalities, such as flat facial profiles, upward-slanting eyes, small ears, short stature, low muscle tone, loose joints, short fingers and toes; and comorbidities, including hypothyroidism and diabetes mellitus [3]. Moreover, there is a high prevalence of people who have co-occurring conditions, and the most frequent co-occurring conditions include vision (72.5%), ear/hearing (71.0%), gastrointestinal (61.3%), respiratory (45.6%), and feeding (33.6%) issues [4]. However, the health problems of people with DS have been systematically monitored and treated, which has led to increased life expectancy. Specifically, in the 1940s, the average life expectancy of people with DS was only 12 years, while newborns are now living for more than 60 years [5]. Thus, further monitoring, understanding, and treating the numerous health conditions and factors that influence health outcomes in DS people are important for further improving their life expectancy and quality of life.

The evaluation of metabolic and physiological health indicators in people with DS is crucial since people with DS are more likely than the general population to suffer from obesity, cardiovascular disease, and metabolic problems [3]. These health indicators include body-built indices, cardiorespiratory fitness, hemoglobin levels, and lipid profiles. First, people with DS are more likely to suffer from metabolic diseases such as insulin resistance, hypercholesterolemia, and inflammation, which are connected to increased BMI and obesity [6]. In addition, the lipid profile, as a strong marker of cardiovascular health and for identifying the risk of heart disease and metabolic disorders, is less favorable in people with DS than in matched peers without DS [7]. Moreover, people with DS generally have decreased levels of physical activity (i.e., fewer than 10% of adults with DS reach the minimum recommended levels of physical activity), which leads to lower cardiorespiratory fitness, which is also one of the strongest health markers [8,9,10]. Indeed, increased cardiorespiratory fitness leads to better metabolic health, enhanced physical functioning, and a decreased risk of cardiovascular disease, which is especially important in people with DS because the most prevalent causes of mortality and morbidity in this population are heart problems [11,12]. Collectively, all these previously mentioned health markers are essential for monitoring the health and well-being of adults with DS and should be systematically included in interventions and programs for improving their health status.

One crucial aspect of the health and well-being of people with DS is the role of their legal guardians. Specifically, since people with DS require lifelong assistance due to their mental state, legal guardians most commonly make decisions regarding health, caregiving, and advocacy [13]. Specifically, a legal guardian’s ability to understand, interpret, and act regarding health information is extremely important for making health decisions and shaping health outcomes in people with DS [14]. Therefore, an important concept called health literacy must be introduced. Health literacy (HL) is most commonly defined as “knowledge, motivation and competences to access, understand, appraise, and apply health information to make judgments and make decisions in everyday life concerning healthcare, disease prevention and health promotion to maintain or improve quality of life during the life course” [15]. HL is considered an important concept which is linked to numerous positive health behaviors across all populations around the globe, including Croatia [15,16,17]. Moreover, low HL is linked to increased hospitalizations and poorer chronic disease outcomes across diverse populations [18]. Indeed, it is well documented that for children with chronic illnesses, there is a direct correlation between parental HL, health behavior, and health outcomes globally [19]. Thus, legal guardians with better HL are supposed to offer more favorable health suggestions and decisions for the well-being of people with DS that they are taking care of [20]. Indeed, it is important to emphasize that delays in seeking medical help, poor adherence to treatment plans, and less than ideal health outcomes for the people under their care might result from guardians’ low HL. One recent study that investigated HL levels in parents of children with DS in Turkey revealed that 81% of the included parents had inadequate and problematic HL levels [14]. However, to date, there is no scientific evidence on the level of HL of parents/legal guardians of adults with DS globally. Furthermore, there are no studies on this topic in Croatia at all, across all age populations of people with DS.

In addition to its clinical relevance, HL also holds ethical and legal significance. According to Article 12 of the United Nations Convention on the Rights of Persons with Disabilities (UNCRPD), individuals with disabilities have the right to participate in decisions affecting their health and well-being on an equal basis with others [21]. To this end, HL is not merely a skill of the caregiver (or parent/legal guardian) but also a significant enabler of supported decision-making [22,23]. Health-literate caregivers are capable of facilitating supportive environments for autonomy and empowering persons with DS to participate effectively in decision-making about their own care. This dual perspective justifies the study’s focus on caregiver HL as both a modifiable determinant of health and a mechanism for promoting autonomy.

Recognizing HL as a determinant of health outcomes globally is growing in recognition; however, evidence of HL in caregivers of adults with intellectual disabilities remains negligible, especially in Southeast Europe. In Croatia, parental HL influences health in adults with DS, especially concerning clinical indicators of health, including metabolic and hematological indices. As our study combines clinical and psychosocial data, we will gather new data that can inform placemaking and care planning, family-centered care, and educational activities designed to meet the needs of especially vulnerable people. Although HL is an important concept for shaping health outcomes, especially in the population with DS, it has not been investigated in detail in the existing scientific literature. Therefore, this study aimed to assess the health status of adults with DS by analyzing their lipid profile, hemoglobin levels, cardiorespiratory fitness, and body-built indices, all stratified by sex. Additionally, this study aimed to evaluate the HL levels of parents/legal guardians and explore the relationship between parental HL levels and the health outcomes of adults with DS. It has to be noted that the novelty of this study is that it is the first study in Croatia that examines the association between parental HL and physiological and functional health outcomes of adults with DS. By identifying areas for intervention, this study aims to support evidence-based care strategies and develop health education programs for parents/legal guardians to promote better health outcomes for adults with DS.

## 2. Materials and Methods

### 2.1. Participants

The participants in this cross-sectional study were 18 people with DS: 11 females and 7 males aged 28.64 ± 9.01 years. All participants were recruited from an NGO that works with people diagnosed with Down syndrome. Additionally, all participants were included in the Nordic walking program at least two times a week. This study was conducted during the month of October 2022, on working days between 08:00 a.m. and 10:00 a.m. The measurements and sampling were performed at the training college for medical students in Split, Croatia. Additionally, we included parents/legal guardians with an average of 53.33 (43–71) years.

This study permitted the inclusion of people with a confirmed diagnosis of Down syndrome (trisomy 21; Q90). The exclusion criteria were reluctance to participate; absence of consent by parents or legal guardians; presence of acute conditions at the time of sampling (e.g., COVID-19 infection); presence of an illness of major organs or organ systems not related to DS; and use of any medication known to affect insulin sensitivity or lipids. The study protocol was approved by the Ethical Board of the University of Split, Faculty of Kinesiology, on 23 September 2021 (EBO: 2181-205-02-01-21-0011). This study was organized and executed according to the principles of the Declaration of Helsinki regarding patient rights.

Successively to ethical approval, the parents or legal guardians of people with DS were presented with the scope of the activity and the procedures to be carried out. Written informed consent was requested to permit people with DS to participate. Primary consent was obtained from legal guardians or parents, who hold legal authority in healthcare and research decisions for the adults with DS included in this study. Informed assent was also asked directly from the adults with DS, using simplified, accessible language and visual aids where appropriate. From a total issuance of 30 consent waivers, 25 recovered, resulting in a response rate of 83%. However, only 18 parents completed the health literacy survey, resulting in 18 dyads (people with DS and their caregivers). Therefore, due to incomplete data from either the participant or the caregiver, only 18 complete dyads were analyzed statistically. At any point during this study, the participants were offered the right to withdraw.

### 2.2. Variables and Measurement

In this study, the following variables were included: level of health literacy (HL) of parents or legal guardians of people with DS; standard anthropometric measures; glucose level; hematological and lipid profiles; and cardiorespiratory fitness of people with DS.

The European Health Literacy Survey Questionnaire (HLS-EU-Q), developed by Sørensen et al. (2013) [24], was used to evaluate the level of HL of parents or legal guardians of people with DS. People were assessed via 47 questions in the questionnaire. The assessment covered an individual’s capacity to collect and comprehend basic health information and to obtain health services. It also assesses the ability of people to use the latter to make appropriate health decisions or to access, comprehend, evaluate, and apply information regarding their health. A general index of HL was created by applying a 4-point Likert scale comprising response options ranging from very difficult (1) to very easy (4). To calculate the score, indexing was applied according to the following formula: index = (mean − 1) × (50/3). Scaling was applied to the HL range, with 0 representing the lowest score and 50 representing the maximum score. The range was then divided into four separate ranking bands as follows: inadequate (from 0–25), problematic (26–33), sufficient (34–42), and excellent (43–50). This study employed the validated Croatian version of the HLS-EU-Q47 questionnaire [25]. To gather the data from the questionnaire, the SurveyMonkey platform was used (SurveyMonkey Inc., San Mateo, CA, USA). Each parent or legal guardian was informed about the procedure and protocols prior to the measurements being taken.

Anthropometric measures included body mass (to the nearest 0.1 kg), body height (standing and sitting to the nearest cm), and body mass index, whereas body-built indices included body mass percentage and muscle mass percentage. The body-built indices were measured via bioelectrical impedance analysis (BIA) (Tanita BC 418 scale; serial number: 15010067, 2015), whereas the remaining variables were measured via a wall-mounted stadiometer for standing height (Holtain, Crymych, UK), with a scale precision of 0.01 cm. The body sitting height variable was measured via the Holtain Harpenden Sitting Height Table SKU: H98607VR (Holtain, Crymych, UK), with a scale precision of 0.5 cm. During the measurements, the participants wore light clothing and were barefoot. The measurements were carried out by trained kinesiology researchers following the standard BIA protocol under strict privacy conditions [26].

Glucose levels and selected hematological parameters, such as hemoglobin and hematocrit, were measured during testing. The participants’ lipid profiles consisted of measurements of triglyceride (TG), total cholesterol (TCHOL), and cholesterol fractional forms. Examples of the latter are high-density lipoprotein (HDL-C), non-high-density lipoprotein–cholesterol (non-HDL-C), low-density lipoproteins (LDL-C), and the CHOL/HDL ratio. Because of the specificity of the population, point-of-care testing (POCT) was used as the diagnostic testing method for evaluating glucose, hemoglobin, hematocrit, and lipid profiles. POCT is a minimally invasive diagnostic method that is widely used because it provides rapid and accurate results [27].

Glucose levels were obtained via a Rightest GM550 (Bionime Corporation, Taichung City, Taiwan) automatic blood glucose analyzer (for home or professional diagnostics) together with the Rightest GS550 test strips. This system employs the reaction between glucose from the capillary blood sample, together with glucose oxidase and potassium ferricyanide (on the diagnostic strip), to obtain the correct glucose reading. The reading is given according to the electrical conductivity of the product of the reaction, which is proportional to the amount of glucose contained in the sample [28].

A Hemochroma PLUS analyzer (Boditech Med Inc., Chuncheon-si, Republic of Korea) was used to collect hematological parameters. Such a device can carry out tests for anemia by employing a disposable test cuvette to take a 15 µL blood sample from a fingertip. A single test takes between 3 and 5 s.

Mission Cholesterol Test Devices and 3-in-1 Lipid Panel Strips (ACON Laboratories, Inc., San Diego, CA, USA) were used to measure the lipid profile. The fresh capillary blood samples for lipid testing were transferred to test strips in the correct volume (35 µL) via capillary transfer tubes from the same producer. The 3-in-1 lipid panel strips directly measure the concentrations of total cholesterol (TCHOL), high-density lipoprotein cholesterol (HDL-C), and triglycerides (TRIG), while the LDL and CHOL/HDL ratios are automatically calculated. In this study, we calculated non-HDL-C because of its potent atherogenic properties compared with those of other lipoproteins [29,30]. Non-HDL-C is calculated as total cholesterol minus HDL-C [31]. High concentrations of non-HDL-C can predict cardiovascular risk according to a multinational study [32].

The devices used during the testing were calibrated through the use of calibration solutions. Manufacturer protocols were followed. A medical doctor supervised the finger punctures required for blood testing and collection of capillary blood drops following the H4-A5 recommendation from The Procedures and Devices for the Collection of Diagnostic Capillary Blood Specimens; Approved Standard—Sixth Edition [33]. The procedure for performing the finger puncture was clearly explained to each parent or legal guardian by a medical doctor prior to taking the sample. This included the recommendation of overnight fasting; information about the procedures, risks, benefits, and rights applicable to the testing; and the ability to ask questions and eliminate doubts.

Most of the studies reported no change in serum TCHOL, LDL-C or HDL-C in people with DS compared with a control group or to population norms [34]. The results were compared with the normal ranges of values for the measured variables recommended by the Croatian Chamber of Medical Biochemists, as shown in Table 1.

The six-minute walk test (6MWT) was used to assess submaximal cardiorespiratory endurance and functional exercise capacity [35]. It was conducted on a flat, straight, 30-m indoor walkway marked with cones at both ends. Prior to the test, each participant was given the same standardized verbal instructions, emphasizing that they should walk at a comfortable pace for six minutes, aiming to cover as much distance as possible without running or jogging. To minimize evaluator bias and variability, the same trained assessor administered all tests, using a stopwatch and standardized script for timing and encouragement (e.g., neutral cues like “You’re doing well” given at one-minute intervals). No physical assistance or pacing was allowed. Environmental conditions, such as lighting, temperature, and noise, were controlled to remain consistent throughout all testing. Only one trial was performed per participant, and the total distance walked was recorded in meters.

### 2.3. Statistical Analysis

The Shapiro-Wilk test was used to determine the normality of the variables. Accordingly, means and standard deviations were reported for all variables.

Differences between males and females in the study’s variables were checked via the *t*-test for independent samples. Additionally, effect sizes (ESs) were calculated and interpreted as follows: <0.02 = trivial; 0.2–0.6 = small; >0.6–1.2 = moderate; >1.2–2.0 = large ES.

The receiver operating characteristic (ROC) curve was calculated between parents with limited (combined inadequate and problematic HL categories) and good (combined sufficient and excellent HL categories) HL levels. The value of the state variable was set as 0 (low HL level) or 1 (good HL level) depending on the thresholding logic, with the aim of obtaining a better-performing model. An area under the curve (AUC) value (a measure of a model’s ability to distinguish between positive and negative classes) above 0.5 was interpreted as significant; an AUC of 0.7 or higher is often considered acceptable, whereas 0.8 or higher is considered good. Furthermore, the relationships between people with DS body-built indices, CRF, lipid profiles, and hemoglobin levels, and their parental/legal guardian’s HL were calculated via Pearson’s correlation coefficients. It has to be noted that only adults with DS who participated in all fitness and blood testing and their respective parents/legal guardians who fulfilled the HL survey were matched and were used in all analyses.

All analyses were performed via Statistica v15 (TIBCO, Palo Alto, CA, USA) with a *p*-value of 0.05.

## 3. Results

Descriptive statistics and sex differences for all the study variables are presented in Figure 1 and are detailed in the Appendix A. There are significantly more males than females with DS, and males have a lower fat mass percentage and a higher muscle mass percentage than females do. Moreover, males have higher hemoglobin and hematocrit values than females do, while females have higher HDL concentrations than males. Finally, males reached better results on the 6MWT.

Findings of gender differences from Figure 1 are further supported by calculated effect sizes, which are presented in Figure 2. Specifically, there was a large ES for body height (Cohen’s D ES 2.75), hematocrit (1.38), HDL (1.75), and the 6MWT (1.36). Moderate ES was found for fat mass percentage (1.12), muscle mass percentage (1.13), glucose (0.86), hemoglobin (1.06), TCHOL (0.62), and LDL (0.73). The rest of the variables displayed small or trivial ES.

Among the 18 parents/legal guardians who filled out the survey, 13 (72.22%) were female. Their overall HL score was 32.36 (ranging from 20.67–47.67). Among the total number of parents/legal guardians, 16.67% were shown to have an inadequate level of HL, 44.4% problematic, 27.77% sufficient, and 11.11% excellent. According to the data, 61.07% of the samples had a limited level of HL.

Table 2 shows the correlation coefficients of the HL subcategories with body indices, lipid profiles, and cardiorespiratory fitness. Total HL is associated with hemoglobin and hematocrit (moderate correlation coefficients). Regarding the HL subcategories, accessing information related to disease prevention, SUB_HC-APPR, SUB_HC-APPL, and SUB_DP-APPL were associated with hematocrit. SUB_DP-U was negatively associated with TCHOL, triglycerides, and non-HDL. Additionally, SUB_DP-APPR was positively associated with the 6 min walk test score.

To evaluate differences in body-built indices, lipid profiles, hematological parameters, and cardiorespiratory fitness between parents with limited and adequate HL status and adults with DS, receiver operating characteristic curves with areas under the curve parameters were calculated and presented in Table 3. The significant variables (AUC above 0.65–0.70) were body height, body mass, hemoglobin, cardiorespiratory fitness, hematocrit, and HDL. However, only hemoglobin and hematocrit reached asymptotic significance.

To better present the results from the ROC curve graphically, the curves for the most significant variables are displayed in Figure 3. Specifically, hematological parameters were the most significant variables that differed between parents with limited and adequate HL status.

## 4. Discussion

There are several main findings of this study. First, the adults with DS included in this study had good lipid statuses and hematological parameters. Second, this study revealed notable sex disparities in health indices among people with DS, with females exhibiting higher levels of HDL and fat mass, whereas males had higher levels of muscle mass, hemoglobin, hematocrit, and body height. Finally, parental HL was related to several health outcomes, with body mass, hemoglobin, hematocrit, 6MWT, and HDL as critical differentiators between persons with DS whose parents had limited versus adequate HL.

### 4.1. Comprehensive Profile of Participants Across Anthropometric, Metabolic, and Functional Variables: Comparisons with Previous Studies and Gender-Based Differences

Our study revealed a high prevalence of obesity or overweight among people living with DS, as shown in other studies. Higher BMI is explained through multiple pathways, from systemic inflammation, reduced muscle tone, lower leptin levels, lower resting energy expenditure, concomitant metabolic diseases, slow metabolism of medications, and poor dietary habits [36,37,38]. As a result, overweight and obesity are prominent phenotypic characteristics of people living with DS [39]. The lower BMI observed in male participants (26.51 vs. 30.10 in females) in this study is likely attributable to their greater height, which naturally distributes body mass more proportionally, as well as potentially greater energy expenditure due to greater muscle mass and metabolic differences. Additionally, our findings revealed that males had a significantly lower fat mass percentage and a higher muscle mass percentage than females did, which may further contribute to their overall lower BMI. These differences align with well-established physiological variations between sexes, where males typically exhibit a more favorable body composition characterized by a higher proportion of lean muscle mass and a lower percentage of body fat [40].

There were notable sex differences in the assessment of exercise capacity (CRF) through the 6MWT, with men scoring significantly higher than women. This is not a surprising result since males often have a greater percentage of muscle mass and may have greater cardiovascular capacity [41]. In DS adults, although both sexes have common limitations of hypotonia, reduced aerobic capacity, and coordination deficits, males may still possess physiological and functional benefits that promote walking performance [42]. Past studies have also shown that DS males exhibit slightly higher physical activity and muscle strength than females, which may influence endurance test results [9,43]. Behavioral factors such as motivation, and self-confidence can also differ by sex and may contribute to variability in performance [44]. These findings suggest that sex should be taken into account when measuring cardiorespiratory fitness in adult persons with DS and are in line with the value of individualized fitness testing and intervention.

Lower levels than the reference values of HDL cholesterol were observed in all participants. However, we found normal values of non-HDL cholesterol, a potential biomarker of metabolic syndrome [45]. Regarding the lipid metabolism of people living with Down syndrome, the literature is not consistent; some studies reported no deviation from normal values in the lipid profile [46,47], and some studies confirmed deviations from normal ranges in the lipidogram values [48,49]. Later, low HDL and elevated triglyceride levels, which represent increased cardiovascular risk, were usually reported. However, some of the studies resulted in lipid profiles within the recommended range for health, although with higher values than those measured in their control group (i.e., siblings) [50]. It is assumed that the alteration in the lipid metabolism profile in some studies is related to lifestyle and dietary habits, especially excess weight and body fat accumulation in the abdominal region [51]. A large body of evidence highlights the findings that people living with DS have unfavorable dietary habits, mainly high in fat and carbohydrates and low in fiber, protein and vitamins [52]. Hypothyroidism, which is common in people living with Down syndrome, can also lead to altered cholesterol profiles, particularly high LDL and low HDL [49]. This may also be associated with a higher reported incidence of diabetes mellitus [36]. Unlike other studies [53], which noted that there were no sex differences in lipid profiles, our female participants had higher HDL concentrations, even though they had higher BMIs, which are usually inversely correlated [54].

In the present study, the glucose level was higher than the reference value, with no sex differences observed. Studies investigating the risk of diabetes mellitus in people with Down syndrome have mixed results. Recent studies reported no differences between subjects with DS and control groups, mostly healthy controls or people with other intellectual disabilities [34,55]. In contrast, several studies have reached different conclusions, with several possible explanations for glucose level impairment in people with DS [56,57]. The greater risk of type 1 diabetes mellitus than in the general population may be the result of extra copies of chromosome 21 and increased defects in the immune system. In fact, in people with DS, the expression of a transcriptional regulator, an autoimmune regulatory protein located on chromosome 21, is reduced. It is reasonable to assume that this may affect the autoimmune reaction in people with DS and the development of type 1 diabetes mellitus. The exact pathways involved are not fully understood, but it is believed that mitochondrial dysfunction and oxidative stress may contribute to increased susceptibility to diabetes mellitus [57]. Several studies have concluded that the prevalence of type 2 diabetes mellitus is greater in people with DS than in the general population, which is assumed to be related to the higher prevalence of obesity among these people [56,57]. However, the observed values are higher than normal but are generally not observed as problematic.

With respect to the hematological profile, we found no deviations from the reference values. A difference was found in higher hemoglobin and hematocrit values in males only. Our results contrast with those of the study by de Gonzalo-Calvo et al. (2020), which resulted in the assessment of hematological parameters among people living with DS with sex taken into account [34]. According to the literature, the impact of trisomy 21 on fetal and postnatal hematopoiesis has been observed and explained through several pathways [58]. As a result, the extra copy of chromosome 21 severely affects the function and number of hematopoietic lineages [59]. Notably, studies have reported lower folate concentrations in people living with DS [34]. Taken together, these findings suggest that trisomy 21 leads to altered hematological parameters due to impaired hematopoiesis, but this study did not confirm these findings.

### 4.2. Health Literacy of Parents/Legal Guardians of Adults with Down Syndrome: Status, Comparisons, and Factors Influencing Differences

The results of this study reveal no difference in the level of HL of parents/legal guardians between people with DS and the general population. The average level of HL in the studied population was 32.36, whereas research on the general population in the Republic of Croatia revealed that the average general HL score was 33.98, which indicates that the level of HL is adequate, although it is on the very border between problematic and adequate [60]. With respect to the level of HL in the general population of EU Member States, three to nearly half of Europeans had low HL [61]. This study revealed that 61.07% (~61%) of parents/legal guardians had a limited level of HL, which is in accordance with a recent study in Turkey; 81% of parents/legal guardians of people with DS had a limited level of HL [14]. This borderline adequacy raises concerns about the potential challenges that parents/legal guardians may face in accessing, understanding, and applying health-related information effectively, particularly when managing the complex healthcare needs of people with DS. Given the critical role that parents/legal guardians play in making healthcare decisions and ensuring adherence to medical and lifestyle recommendations for people with DS, these results emphasize the need for targeted interventions to enhance HL among parents and legal guardians.

This study revealed that the total level of HL, together with subcategories/domains of health care and disease prevention, are positively correlated with the hematological fractions (hemoglobin and hematocrit). According to the WHO definition, anemia in adults is a state of reduced erythrocytes or hemoglobin (Hb) [62]. Anemias are very common diseases: total cases of anemia increased from 1.42 (1.41–1.43) billion in 1990 to 1.74 (1.72–1.76) billion in 2019; one in four people globally has some form of anemia [63]. It is estimated that anemia affects half a billion women aged 15–49 years worldwide [62]. A recent publication revealed that women of reproductive age from urban areas and with some form of tertiary education had good knowledge (causes, consequences, prevention and management) of anemia [64]. Furthermore, the same study revealed that awareness of anemia was increased through media, school, and education programs from professional medics. These findings explain the positive correlation of parental HL with hematological fractions in this study. Specifically, most of the participants in this study were female (13 of them) living in the second largest city in Croatia. Additionally, most of them had a secondary school level of education, so it can be hypothesized that living in an urban community with easy access to information and a health care system can be connected with good knowledge of anemia and therefore act as a preventative factor. This suggests that higher HL among parents/legal guardians may lead to better nutritional and medical management, ultimately benefiting the hematological health of people with DS under their care.

The present study revealed a negative association between parental/legal guardians’ HL (subcategory disease prevention) and fractions of lipograms, such as TCHOL, TRIG, and non-HDL. As mentioned previously (please see the first part of the Discussion), people with DS in this study had normal lipid values (except for lower HDL values), whereas the HL level of parents/legal guardians was problematic (the average level of the sample was 32.36). Inadequate HL is repeatedly reported to be linked with adverse lipid profiles and poor health outcomes [17,19,65], which makes the abovementioned results uncommon and controversial. One explanation for such findings could be that something other than the parental level of HL influences lipid values in adults with DS. All participants in this research were members of a local nongovernmental organization (NGO). Perhaps the NGO, through its mission of providing adequate care for people with DS, made people more aware of a healthy lifestyle and dietary adaptation through various educational programs and workshops, generally promoting healthy habits. Split, the second largest city in Croatia, is situated on the Adriatic coast, and traditionally, cuisine is based on a Mediterranean diet. Recently, the positive effects of the Mediterranean diet on lipid metabolism in children with DS were demonstrated, which could explain the relatively good lipid profile [48]. Additionally, people with DS also have a high incidence of thyroid dysfunctions, which impacts lipid processing [49]. All the abovementioned factors may have influenced the results of this study. However, dietary habits and thyroid parameters were not within the scope of the study but should be further investigated in future studies.

Another positive correlation obtained in the research was between parental/legal guardian levels of HL (subcategory disease prevention) and the 6MWT. Multiple studies have confirmed a positive correlation between parental HL and disease prevention, which includes physical activity. Specifically, in a systematic review of 35 studies, it was concluded that adults with DS can accrue health benefits from properly designed physical activity and exercise interventions [42]. Although the parents’/legal guardians’ HL levels were inadequate, the results of the 6MWT of adults with DS are good, although people with DS tend to have a lower physical activity level due to muscle hypotonia [66]. One possible reason for this result is that, even though parental or legal guardian HL levels have been found to be inadequate in some areas, they may have higher-than-measured HL in particular, which is a useful aspect of disease prevention, especially in regard to encouraging and supporting physical activity. Additionally, parents and guardians may be able to encourage a more active lifestyle for people with DS through lived experience, informal knowledge acquisition, and access to community services or healthcare experts. The precise processes by which parental HL affects physical activity engagement should be investigated further, as should the ways in which focused interventions may improve parental HL and the health outcomes that follow for people with DS.

The present results are consistent with international evidence that underscores the important contribution of parental HL to managing chronic conditions, as well as enhancing the quality of life of people with intellectual disabilities, including DS. Research from the United States and Western Europe has similarly found that higher caregiver HL status is correlated with better adherence to medical recommendations, better preventive care and fewer hospitalizations [67,68]. In line with trends in literature, our study found positive associations between higher parental HL and better metabolic and functional outcomes in adults with DS [14]. However, in contrast to much of the literature, which has historically focused on children, our study imparts information on the role of HL in adults, which has been less represented in this context. While studies often look at HL in broader disability groups, this study is able to narrowly identify the DS population in a Croatian context, which adds new, uncharted information on the DS population. The other finding of note was that there were no significant gender differences across various domains, which is different from findings among larger international cohorts who found that females with DS sometimes have different health trajectories compared with males with DS [69]. Differences such as these may point to cultural, systemic, or sample differences, which may warrant follow-up cross-cultural studies.

These findings also have important rights-based implications. Adults with Down syndrome have the right, under Article 12 of the UN Convention on the Rights of Persons with Disabilities (UNCRPD), to participate in health decisions with appropriate support [21]. In this context, caregiver HL should not only inform management of health but also enable supported decision-making, respecting the autonomy and preferences of people with DS. Enhancing HL among caregivers may, therefore, help towards more communicative inclusiveness and enable individuals with DS to better manage their own care.

Overall, the results highlight how optimizing parental HL may facilitate support of the health of adults with DS. The findings of this study have important implications for public health policy and caregiver education. The observed associations between parental HL and clinical outcomes, particularly hematological parameters and functional fitness, suggest that caregiver knowledge is a modifiable factor influencing the health of adults with DS. These insights highlight the potential of targeted, evidence-based educational interventions as part of public health strategies. Possible tactics to improve parental HL in these settings may include conducting HL workshops for caregivers and parents based on their various levels of literacy, routinely screening for HL in healthcare departments, and developing co-designed, user-friendly educational materials (e.g., visual guides, simplified instructions). Peer mentoring approaches that connect experienced caregivers with less experienced caregivers for informal learning can also help strengthen the informal learning networks. More importantly, engagement of interdisciplinary teams made up of healthcare providers, informal caregivers, educators, and policy makers can help institutionalize HL promotion as part of broader care systems. Investment in these types of practices can encourage families’ capacity to manage medical complexity, improve continuity of care, and improve health outcomes for people with DS.

### 4.3. Limitations of the Study

The main limitation of this study is the relatively small sample size (18 participants with DS and 18 legal guardians), which limits the generalizability of the findings, particularly with respect to gender differences and the influence of parental HL on health markers. The cross-sectional study design restricts the ability to establish causal relationships between HL and health outcomes. Although POCT is beneficial for compliance, it is also possibly less accurate than traditional venous blood collection, which may affect the validity of lipid and hematological assays. In addition, the analytical sensitivity of POCT is poor, and the potential variation between POCT devices and laboratory assays needs to be considered when results are being interpreted, especially in studies with small sample sizes where differences in measurements may be exaggerated. Additionally, the study did not account for important lifestyle factors such as dietary habits, thyroid function, or overall physical activity levels outside the Nordic walking program, all of which could influence metabolic health. The HL assessment relied on a self-report questionnaire, which may introduce response bias and may not fully capture parents’/legal guardians’ practical ability to make informed health decisions. Additionally, we included a relatively homogeneous sample of people with DS who were able to be included in the Nordic walking program. Indeed, the majority of people with DS have comorbidities and a low level of mobility, which usually limits their participation in such exercise programs. Thus, future studies should include a more heterogeneous sample to support the results. Given the limited sample size and cross-sectional design, these findings should be considered exploratory and interpreted with caution.

Despite these limitations, this study has several notable strengths. It provides a comprehensive health assessment of adults with DS, evaluating a wide range of metabolic, physiological, and functional health markers, including lipid profiles, hematological parameters, cardiorespiratory fitness, and body composition. A key strength is the inclusion of parental HL, highlighting its potential impact on health outcomes in people with DS, which is an area that remains largely underexplored. Additionally, the study conducted gender-stratified analyses, offering valuable insights into gender-based physiological differences in this population. Another strength is the standardized Nordic walking program, which ensured that all participants had a comparable level of physical activity, reducing variability in fitness outcomes. Moreover, the use of point-of-care testing for blood sampling improved participant compliance and reduced discomfort, making it a more feasible method for people with DS who may have difficulties with traditional venous blood collection.

## 5. Conclusions

Important information about the health status of adults with DS and how parental HL influences their health outcomes is provided by this study. The results reveal that the participants generally had acceptable lipid profiles, hematological markers, and cardiorespiratory fitness despite a high prevalence of overweight and obesity. This was probably due to their participation in an organized Nordic walking program. Compared with females, males had greater muscle mass, lower fat mass, better CRF, and higher hemoglobin and hematocrit levels, indicating significant sex differences. Crucially, parental HL was linked to important health indicators, specifically functional fitness and hematological parameters, highlighting the possible influence of their knowledge on the well-being of people with DS.

This study uniquely contributes to the literature by empirically linking parental HL with objective health indicators in adults with DS, a topic rarely addressed in Southeast European settings. Given the lack of data from Croatia, our findings highlight the need for greater investment in caregiver and parental education and systemic support to optimize health outcomes in adults with DS. Specifically, public health authorities and governing bodies should develop and promote educational materials specifically designed for improving HL of parents and legal guardians responsible for children or adults with DS or other disabilities. Future research should use longitudinal designs to examine the causal links between HL and health outcomes, including more metabolic and lifestyle factors, and look into focused interventions meant to improve HL in carers. The sampling process should also be broadened to larger, more diverse populations within different socio-economic contexts and locations to promote generalizability. Finally, longitudinal research should consider the implementation of targeted specific interventions for HL, such as legal guardians’ education, specific digital devices, or community-based workshops, and study the improvements of HL on health outcomes (e.g., cardiovascular fitness or functional independence) within a longitudinal design. Improving the HL of parents and legal guardians may be crucial for maximizing illness prevention and health promotion for people with DS, which will eventually improve their long-term health and quality of life.

## Figures and Tables

**Figure 1 healthcare-13-01212-f001:**
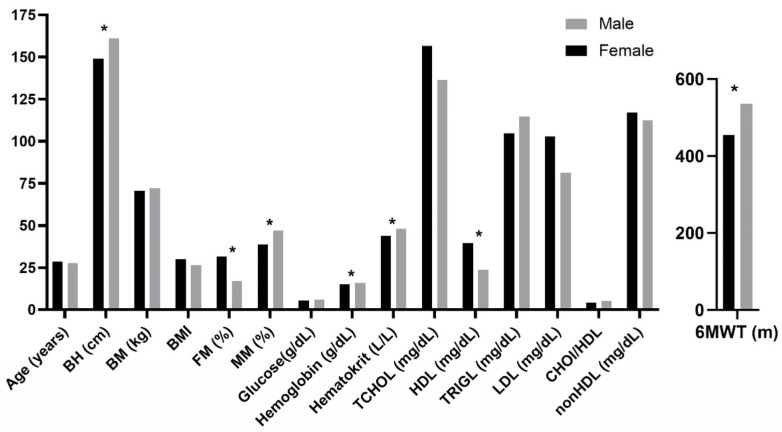
Gender differences in all study variables. Note: BH—body height; BM—body mass; BMI—body mass index; FM%—fat mass percentage; MM%—muscle mass percentage; TCHOL—total cholesterol; HDL—high-density lipoprotein; TRIGL—triglycerides; LDL—low-density lipoprotein; CHOL/HDL—cholesterol to HDL ratio; nonHDL—non-HDL cholesterol; 6MWT—six-minute walk test * *p* < 0.05.

**Figure 2 healthcare-13-01212-f002:**
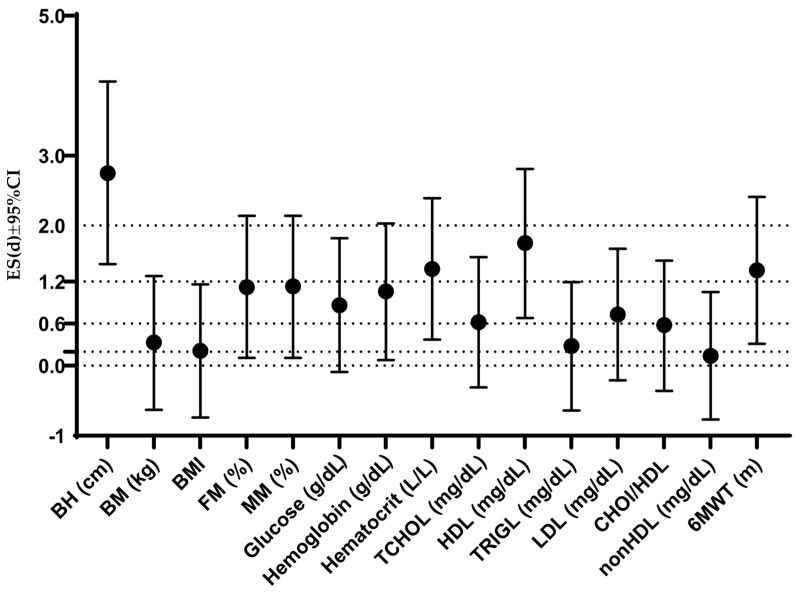
Effect sizes and 95% confidence intervals for gender differences in all study variables. BH—body height; BM—body mass; BMI—body mass index; FM%—fat mass percentage; MM%—muscle mass percentage; TCHOL—total cholesterol; HDL—high-density lipoprotein; TRIGL—triglycerides; LDL—low-density lipoprotein; CHOL/HDL—cholesterol to HDL ratio; nonHDL—non-HDL cholesterol; 6MWT—six-minute walk test.

**Figure 3 healthcare-13-01212-f003:**
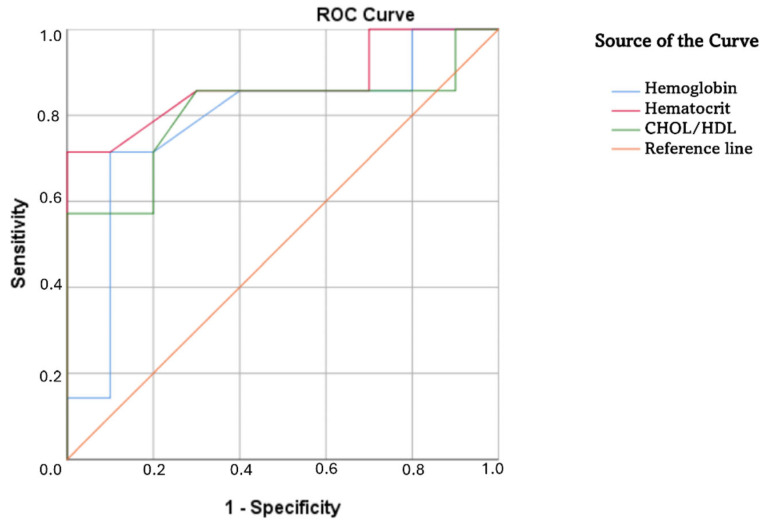
Receiver operating characteristic curve for differences between parental HL status for the most significant variables. Note: CHOL/HDL—cholesterol-to-high-density lipoprotein ratio.

**Table 1 healthcare-13-01212-t001:** The normal ranges of values for hemoglobin, hematocrit, glucose, cholesterol, HDL-C, LDL-C, and triglycerides recommended by the Croatian Chamber of Medical Biochemists.

Analyst/Search/Index	Gender	Years	Reference Interval
**Hemoglobin**	male	≥20	13.8–17.5 g/dL
	female	≥20	11.9–15.7 g/dL
**Hematocrit**	male	≥20	41.5–53.0 dL/dL
	female	≥20	35.6–47.0 dL/dL
**Glucose**	male, female	20–30	59.4594–93.6936 mg/dL
	male, female	>30	63.063–100.900 mg/dL
**Cholesterol**	Male, female	adult	<193.35 mg/dL
**HDL-cholesterol**	male	adult	>38.67 mg/dL
	female	adult	>46.40 mg/dL
**LDL-cholesterol**	male, female	adult	<116.01 mg/dL
**Triglycerides**	male, female	adult	<150.56 mg/dL
**Non-HDL**	male, female	adult	<130 mg/dL

**Table 2 healthcare-13-01212-t002:** Correlations between health literacy and body indices, lipid profiles, and cardiorespiratory fitness.

Variable	HL-T	HL-1	HL-2	HL-3	HL-4	HL-5	HL-6	HL-7	HL-8	HL-9	HL-10	HL-11	HL-12
**Age**	−0.03	−0.08	0.02	−0.10	0.25	0.20	−0.11	−0.18	−0.30	0.11	−0.21	−0.19	0.10
**BH**	0.17	0.31	0.45	0.21	0.06	−0.01	0.13	0.21	0.32	−0.34	0.20	0.37	−0.38
**BM**	0.31	0.43	0.45	0.08	0.24	0.33	0.25	−0.07	0.22	0.15	0.40	0.29	−0.12
**BMI**	0.26	0.30	0.25	0.03	0.31	0.45	0.19	−0.21	0.07	0.28	0.29	0.11	0.10
**FM%**	0.08	0.01	−0.05	−0.01	0.18	0.28	0.17	−0.35	0.01	0.21	0.18	−0.16	0.19
**MM%**	−0.08	−0.01	0.05	0.01	−0.18	−0.28	−0.17	0.35	−0.01	−0.21	−0.18	0.16	−0.20
**Glucose**	0.22	0.36	0.35	0.17	0.00	0.21	0.09	0.11	0.09	0.15	0.11	0.32	−0.04
**Hemoglobin**	0.46	0.37	0.43	0.29	0.04	0.07	0.03	0.43	0.57 **	0.39	0.34	0.47 *	0.21
**Hematocrit**	0.47 *	0.52 *	0.56 **	0.38	0.48 *	0.11	0.16	0.29	0.54 *	0.12	0.31	0.37	−0.07
**TCHOL**	−0.19	−0.28	−0.20	−0.17	−0.19	0.03	−0.46 *	−0.23	−0.26	0.14	−0.36	−0.27	0.39
**HDL**	−0.32	−0.40	−0.52 *	−0.12	−0.15	−0.18	−0.14	−0.14	−0.37	−0.05	−0.25	−0.34	0.20
**TRIGL**	0.04	−0.17	−0.11	−0.24	−0.16	0.04	−0.49 *	0.21	0.08	0.38	−0.09	0.08	0.46
**LDL**	−0.16	−0.15	−0.17	−0.09	−0.06	−0.11	−0.34	−0.12	−0.34	0.08	−0.29	−0.20	0.20
**CHOl/HDL**	0.38	0.31	0.43	0.15	0.15	0.13	−0.19	0.31	0.23	0.33	0.09	0.31	0.13
**nonHDL**	−0.09	−0.15	−0.01	−0.14	−0.15	0.11	−0.46 *	−0.20	−0.16	0.18	−0.32	−0.18	0.36
**6MWT**	0.16	0.06	0.23	0.26	−0.15	−0.27	0.01	0.50 *	0.45	−0.01	0.14	0.23	−0.07

Note: * *p* < 0.05; ** denotes *p* < 0.01; BH—body height; BM—body mass; BMI—body mass index; FM%—fat mass percentage; MM%—muscle mass percentage; Glucose—blood glucose level; Hemoglobin—hemoglobin concentration; Hematocrit—hematocrit percentage; TCHOL—total cholesterol; HDL—high-density lipoprotein; TRIGL—triglycerides; LDL—low-density lipoprotein; CHOL/HDL—cholesterol to HDL ratio; nonHDL—non-HDL cholesterol; 6MWT—six-minute walk test; HL-T—health literacy total score; HL-1—accessing healthcare-related information; HL-2—understanding healthcare-related information; HL-3—appraising healthcare-related information; HL-4—applying healthcare-related information; HL-5—accessing information related to disease prevention; HL-6—understanding information related to disease prevention; HL-7—appraising information related to disease prevention; HL-8—applying information related to disease prevention; HL-9—accessing information related to health promotion; HL-10—understanding information on health promotion; HL-11—appraising in-formation on health promotion; HL-12—applying information on health promotion.

**Table 3 healthcare-13-01212-t003:** Differences between parents with low and adequate health literacy levels in all study variables of people with DS expressed through receiver operating characteristic values.

Variables	AUC	SE	AS	95% CI
**BH**	0.70	0.16	0.19	0.39–1.00
**BM**	0.72	0.13	0.16	0.46–0.97
**BMI**	0.62	0.15	0.45	0.32–0.91
**FM%**	0.48	0.17	0.91	0.15–0.82
**Glucose**	0.61	0.15	0.46	0.32–0.89
**Hemoglobin**	0.79	0.12	0.05	0.54–1.00
**Hematocrit**	0.87	0.10	0.01	0.68–1.00
**TCHOL**	0.61	0.14	0.46	0.33–0.89
**TRIGL**	0.32	0.13	0.22	0.06–0.58
**LDL**	0.64	0.15	0.33	0.36–0.93
**nonHDL**	0.51	0.15	0.96	0.22–0.80
**HDL**	0.71	0.13	0.16	0.45–0.96
**6MWT**	0.67	0.14	0.24	0.39–0.95

Note: BH—body height; BM—body mass; BMI—body mass index; FM%—fat mass percentage; glucose—blood glucose level; hemoglobin—hemoglobin concentration; hematocrit—hematocrit percentage; TCHOL—total cholesterol; HDL—high-density lipoprotein; TRIGL—triglycerides; LDL—low-density lipoprotein; nonHDL—non-HDL cholesterol; 6MWT—six-minute walk test; AUC—area under the curve; SE—standard error; AS—asymptotic significance; CI—confidence interval.

## Data Availability

Data are available upon reasonable request.

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
