# Peer review of "Metabolic, Hematological, and Functional Health in Adults with Down Syndrome and Significance of Parental Health Literacy: A Cross-Sectional Study"

_healthcare, 2025, doi:10.3390/healthcare13101212_

Round 1

Reviewer 1 Report (Previous Reviewer 1)

Comments and Suggestions for Authors

Dear authors,
Congratulations on the excellent topic and research. The changes suggested earlier have been duly rectified and/or responded to. However, I suggest making some minor changes:
Abstract: line 18 gaurdians change to guardians
Study limitations: line 564 "small sample size (18 participants)," only 18 participants or (18+18 participants)?
In the supplementary file still keep the 25 participants with DS see (Total sample (n=25))- please rectify to n=18.

Author Response

Dear authors,
Congratulations on the excellent topic and research. The changes suggested earlier have been duly rectified and/or responded to. However, I suggest making some minor changes:

RESPONSE: Thank you very much for the support and for valuing our work.

Abstract: line 18 gaurdians change to guardians

RESPONSE: Amended accordingly.

Study limitations: line 564 "small sample size (18 participants)," only 18 participants or (18+18 participants)?

RESPONSE: It is 18 plus 18, so total 36.

In the supplementary file still keep the 25 participants with DS see (Total sample (n=25))- please rectify to n=18.

RESPONSE: Thank you for noticing, it is now corrected.

Reviewer 2 Report (Previous Reviewer 2)

Comments and Suggestions for Authors

Reviewer Comments:

Although the manuscript has improved formally, the authors should: restructure the introduction and discussion to reduce density and redundancies, and better integrate the practical implications with the data actually obtained.

1.Abstract

- A better presentation of the objective, the sample and the main variables has been included.

- The clinical impact of the findings is still vague. The sentence “acceptable health indicators can be explained by...” does not make it clear why it is clinically relevant.

- The conclusion lacks force. It should more directly link parental LH with practical implications.

2. Introduction

- The part on human rights (UNCRPD) is interesting, but it breaks the logical flow of the clinical and methodological approach if it is not better articulated with the rest of the text.

3. Methods

- The inclusion of tools such as POCT is valid, but the discussion on their validity should also be under “Limitations”.

4. Results

- Emphasize that the findings obtained are exploratory and should be interpreted with caution.

Author Response

Although the manuscript has improved formally, the authors should: restructure the introduction and discussion to reduce density and redundancies, and better integrate the practical implications with the data actually obtained.

RESPONSE: We revised the Introduction and Discussion to reduce length and remove redundancies, ensuring a clearer and more focused narrative.

Practical implications are now more directly tied to the findings, particularly the links between parental HL and hematological/functional outcomes, highlighting opportunities for targeted caregiver education.

Please see improvements throughout the text.

1.Abstract

- A better presentation of the objective, the sample and the main variables has been included.

- The clinical impact of the findings is still vague. The sentence “acceptable health indicators can be explained by...” does not make it clear why it is clinically relevant.

RESPONSE: Thank you for this important observation. We revised the Abstract to clarify the clinical relevance of the findings. Specifically, we replaced the sentence in question with a clearer statement highlighting the significance of the results:

“These findings suggest that in the absence of severe comorbidities, adults with DS may achieve stable health profiles, particularly when supported by structured physical activity and informed caregiving.” This revision aims to more clearly express the clinical implications regarding screening, health maintenance, and the role of caregiver education.

- The conclusion lacks force. It should more directly link parental HL with practical implications.

RESPONSE: It is now, we believe, improved. Text reads: “Thus, our findings highlight the need for greater investment in caregiver and parental health  education and systemic support to optimize health outcomes in adults with DS.”

  1. Introduction

- The part on human rights (UNCRPD) is interesting, but it breaks the logical flow of the clinical and methodological approach if it is not better articulated with the rest of the text.

RESPONSE: We tried to articulate that paragraph better, text now reads: “In addition to its clinical relevance, HL also holds ethical and legal significance. According to Article 12 of the United Nations Convention on the Rights of Persons with Disabilities (UNCRPD), individuals with disabilities have the right to participate in decisions affecting their health and wellbeing on an equal basis with others [21]. To this end, HL is not merely a skill of the caregiver (or parent/legal guardian) but also a sig-nificant enabler of supported decision-making [22,23]. Health-literate caregivers are capable of facilitating supportive environments for autonomy and empowering persons with DS to participate effectively in decision-making about their own care. This dual perspective justifies the study’s focus on caregiver HL as both a modifiable determinant of health and a mechanism for promoting autonomy.”

  1. Methods

- The inclusion of tools such as POCT is valid, but the discussion on their validity should also be under “Limitations”.

RESPONSE: Thank you for this important observation. In response, we have added a specific note under the Limitations section addressing the potential limitations of using point-of-care testing (POCT). While POCT offers practical advantages in this population (e.g., feasibility and compliance), we now acknowledge its lower accuracy compared to standard laboratory methods and the potential for measurement variability. This clarification reinforces transparency about methodological choices and data interpretation.

Text reads: Although POCT is beneficial for compliance, it is also possibly less accurate than traditional venous blood collection is, which may affect the validity of lipid and hematological assays. In addition, the analytical sensitivity of POCT is poor, and the potential variation between POCT devices and laboratory assays needs to be considered when results are being interpreted, especially in studies with small sample sizes where differences in measurements may be exaggerated.

  1. Results

- Emphasize that the findings obtained are exploratory and should be interpreted with caution.

RESPONSE: We appreciate this important point. To address it, we have added a clarifying statement in the Limitation section to emphasize that the findings are exploratory and should be interpreted with caution, especially given the small sample size and cross-sectional design. Text reads: ‘Given the limited sample size and cross-sectional design, these findings should be considered exploratory and interpreted with caution. Further studies with larger and more diverse populations are necessary to confirm these associations.’

Reviewer 3 Report (Previous Reviewer 3)

Comments and Suggestions for Authors

The authors response well.

Author Response

Comment: The authors response well.

Response: Thank you for supporting our work!

This manuscript is a resubmission of an earlier submission. The following is a list of the peer review reports and author responses from that submission.

Round 1

Reviewer 1 Report

Comments and Suggestions for Authors

The study is fascinating and brings novelty. However, significant ethical and methodological limitations need to be reviewed. One concerns the autonomy and rights of adults with DS, both in terms of their consent and their right to be informed. A major methodological limitation is the fact that the sample includes 25 adults with DS (which are identified as children in Table 4?), but with only 18 HL questionnaires completed by parents — how does this correlate? The sample consists of 18 adults with DS and 18 parents or legal guardians, so you should correlate 18 and not 25 parameters for adults with DS. Due to these issues, I have not reviewed it from the discussion onwards.

Author Response

Reviewer 1

The study is fascinating and brings novelty. However, significant ethical and methodological limitations need to be reviewed. One concerns the autonomy and rights of adults with DS, both in terms of their consent and their right to be informed. A major methodological limitation is the fact that the sample includes 25 adults with DS (which are identified as children in Table 4?), but with only 18 HL questionnaires completed by parents — how does this correlate? The sample consists of 18 adults with DS and 18 parents or legal guardians, so you should correlate 18 and not 25 parameters for adults with DS. Due to these issues, I have not reviewed it from the discussion onwards.

RESPONSE: We sincerely thank the reviewer for the thoughtful and constructive feedback. We acknowledge the importance of ethical rigour and methodological clarity, especially when working with vulnerable populations such as adults with Down syndrome.

We greatly appreciate the reviewer’s emphasis on the ethical considerations surrounding the autonomy and rights of adults with Down syndrome. We fully recognize that, regardless of cognitive ability, adults with DS are individuals with legal and moral rights to dignity, information, and participation in research. In this study, informed consent procedures followed the ethical standards outlined in the Declaration of Helsinki and were approved by the Institutional Ethics Committee. Due to varying levels of cognitive and legal capacity among participants, we adopted a dual-consent approach: Primary consent was obtained from legal guardians or parents, who hold legal authority in healthcare and research decisions for the adults with DS included in this study. Informed assent was also sought directly from the adults with DS, using simplified, accessible language and visual aids where appropriate. Participants were given the opportunity to ask questions and were included in decision-making to the greatest extent possible, in line with supported decision-making principles. This approach ensured that adults with DS were not excluded from the ethical process, but rather involved to the fullest extent that was ethically and cognitively appropriate, respecting both their autonomy and their need for protection. We have clarified this process further in the revised Methods section to ensure transparency and compliance with ethical norms. Text now reads: “Written, informed consent was requested to permit individuals with DS to participate. Primary consent was obtained from legal guardians or parents, who hold legal authority in healthcare and research decisions for the adults with DS included in this study. Informed assent was also asked directly from the adults with DS, using simplified, accessible language and visual aids where appropriate.”

We acknowledge the confusion regarding the sample size. The initial pool included 25 adults with DS. However, only 18 pairs of adults with DS and their respective caregivers completed both the clinical assessments and the Health Literacy Questionnaire. Therefore, we have revised the manuscript to clearly indicate that only data from the 18 matched pairs were used in correlation and comparative analyses.

Thank you for pointing out the inconsistency in terminology. The use of the term “children” in Table 4 was an editorial oversight. It has been corrected to “adults with DS” throughout the manuscript to ensure accurate representation and respect for participant identity.

Reviewer 2 Report

Comments and Suggestions for Authors
  1. Title
  • Indicate the type of study
  1. Abstract
  • clarify and indicate the research problem in order to be able to address the study objective
  • Better explain the clinical impact of the findings.
  1. Introduction
  • It is important to emphasize why this study is novel and to address research gaps.
  • Some parts are dense, the flow could be improved with shorter paragraphs, to better understand the information.
  1. Methods
  • Explain in more detail what was taken into account when performing the 6-minute walk test and how the procedure was carried out to avoid bias in the evaluation.

     5. Dicussion and Conclusion

  • The explanation of the implications of parental health literacy on the health of adults with DS could be strengthened.
  • More concrete suggestions for future research could be included.

Author Response

Reviewer 2

  1. Title
  • Indicate the type of study

RESPONSE: It is now clarified; new title is : Metabolic, Hematological, and Functional Health in Adults with Down Syndrome and Significance of Parental Health Literacy; A Cross-sectional Study

  1. Abstract
  • clarify and indicate the research problem in order to be able to address the study objective
  • Better explain the clinical impact of the findings.

RESPONSE: Thank you for these suggestions, it is now clarified and added throughout the abstract.

  1. Introduction
  • It is important to emphasize why this study is novel and to address research gaps.
  • Some parts are dense, the flow could be improved with shorter paragraphs, to better understand the information.

RESPONSE: Thank you for suggesting this. The explanataion about the novelty and what are the research gaps is now written in the Introduction section, text reads: “Recognizing HL as a determinant of health outcomes globally is growing in recognition; however, evidence of HL in caregivers of adults with intellectual disabili-ties remains negligible, especially in Southeast Europe. In Croatia, parental HL influ-ences health in adults with, especially concerning clinical indicators of health including metabolic and hematological indices. As our study combines clinical and psychosocial data, we will gather new data that can inform placemaking and care planning, fami-ly-centered care, and educational activities designed to meet the needs of especially vulnerable individuals.Although HL is an important concept for shaping health out-comes, especially in the population with DS, it has not been investigated in detail in the existing scientific literature. Therefore, this study aimed to assess the health status of adults with DS by analyzing their lipid profile, hemoglobin levels, cardiorespiratory fitness, and body-built indices, all of which were stratified by sex. Additionally, this study aimed to evaluate the HL levels of parents/legal guardians and explore the rela-tionship between parental HL levels and the health outcomes of adults with DS. It has to be noted that the novelty of this study is that it is the first study in Croatia that examines the association between parental HL and physiological and functional health outcomes of adults with DS.”

  1. Methods
  • Explain in more detail what was taken into account when performing the 6-minute walk test and how the procedure was carried out to avoid bias in the evaluation.

RESPONSE: Thank you for your thoughtful comment. We agree that providing more detail on the procedure enhances transparency and strengthens methodological rigor. To address this, we have revised the Methods section to clarify how the 6MWT was standardized to minimize potential sources of bias. Specifically, we ensured that all participants received identical verbal instructions and encouragement, the test environment was kept consistent for all sessions (quiet hallway, same time of day), and a single trained evaluator conducted all assessments to reduce inter-rater variability. These steps were implemented to ensure that the distance walked reflected participants' actual functional capacity without external influence or variability in administration.

  1. Dicussion and Conclusion
  • The explanation of the implications of parental health literacy on the health of adults with DS could be strengthened.

RESPONSE: It is now explained in detail in the Discussion section, please see the following text: “The present results are consistent with international evidence that underscores the important contribution of parental HL to managing chronic conditions, as well as en-hancing quality of life, of people with intellectual disabilities, including DS. Research from the United States and Western Europe has similarly found that higher caregiver HL status is correlated with better adherence to medical recommendations, better preventive care and fewer hospitalizations [65,66]. In line with trends in literature, our study found positive associations between higher parental HL and better metabolic and functional outcomes in adults DS [14]. However, in contrast to much of the literature which has historically focused on children, our study imparts information on the role of HL in adults, which has been less represented in this context. While studies often look at HL in broader disability groups, this study is able to narrowly identify the DS popula-tion in a Croatian context, which adds new uncharted information on the DS population. The other finding of note was that there were no significant gender differences across various domains, which is different from findings among larger international cohorts who found that females with DS sometimes have different health trajectories compared to males with DS [67]. Differences such as these may point to cultural, systemic or sample differences, which may warrant follow-up cross-cultural studies.”

  • More concrete suggestions for future research could be included.

RESPONSE: Thank you for this helpful suggestion. In response, we have expanded the concluding section of the manuscript to include specific and actionable recommendations for future research. Text reads: “The sampling process should also broaden to larger, more diverse populations within different socio-economic contexts and locations to promote generalizability. Finally, longitudinal research should consider the implementation of targeted specific inter-ventions for HL such as legal guardian’s education, specific digital devices, or community-based workshops, and study the improvements of HL on health outcomes (e.g., cardiovascular fitness or functional independence) within a longitudinal design.”

Reviewer 3 Report

Comments and Suggestions for Authors

The article titled "Metabolic, Hematological, and Functional Health in Adults with Down Syndrome: The Significance of Parental Health Literacy" provides valuable insights into the health status of adults with Down syndrome (DS) and the impact of parental health literacy (HL) on their health outcomes. While the study is well-conducted and presents significant findings, there are areas where minor revisions could enhance clarity, organization, and impact.

  1. The abstract is informative but could benefit from more concise language. Consider reducing repetition and focusing on key results and implications.
  2. Expand on the significance of HL globally to provide broader context before narrowing down to Croatia.

  3. Clarify why this study uniquely contributes to existing knowledge, especially given the limited research in Croatia.

  4. While results are clear, consider reorganizing tables or figures to make gender differences and HL correlations more visually accessible.

  5. Elaborate on how findings align or contrast with previous studies globally.

  6. Discuss practical recommendations for improving parental HL further, including specific intervention strategies

Author Response

Reviewer 3

The article titled "Metabolic, Hematological, and Functional Health in Adults with Down Syndrome: The Significance of Parental Health Literacy" provides valuable insights into the health status of adults with Down syndrome (DS) and the impact of parental health literacy (HL) on their health outcomes. While the study is well-conducted and presents significant findings, there are areas where minor revisions could enhance clarity, organization, and impact.

  1. The abstract is informative but could benefit from more concise language. Consider reducing repetition and focusing on key results and implications.

RESPONSE: Thank you for this suggestion. The abstract is now simplified.

  1. Expand on the significance of HL globally to provide broader context before narrowing down to Croatia.

RESPONSE: I appreciate your useful comment. We also believe that integrating a broader context into the importance of HL will improve the introduction of the manuscript. We, therefore, included evidence that highlights the impact of HL across various health systems and populations within different countries. This approach will better integrate the focus on Croatia and improve the international relevance of the study. Please see corrections throughout the Introduction.

  1. Clarify why this study uniquely contributes to existing knowledge, especially given the limited research in Croatia.

RESPONSE: Thank you for highlighting the importance of clarifying our study’s contribution. We agree that further emphasizing its novelty and relevance is crucial, particularly within the Croatian context. This study is, to the best of our knowledge, the first in Croatia to examine the relationship between parental HL and the metabolic, hematological, and functional health status of adults with DS. While international literature has explored the role of HL in health outcomes, few studies have focused on familial HL as a determinant of clinical health markers in adults with intellectual disabilities — and virtually none within Southeast European populations.

Moreover, Croatia currently lacks population-specific data on both HL levels among caregivers of individuals with DS and their influence on health management. This study contributes novel empirical evidence to fill that gap, offering insight into how non-clinical parental factors (such as HL) may affect the physiological well-being of adults with DS. By addressing this under-researched intersection between HL, caregiver roles, and clinical outcomes in adults with DS, our work provides a foundation for policy-level discussions, caregiver education strategies, and targeted interventions within the Croatian healthcare and social care systems.

These points have now been emphasized in the Introduction and Discussion sections to better position the study within both the national and international research landscapes.

  1. While results are clear, consider reorganizing tables or figures to make gender differences and HL correlations more visually accessible.

RESPONSE: Thank you for this suggestion. We added Figure 1 instead of Table 2 and added Supplementary table 1 with detailed results. Also, we simplified Table 3 and believe it is now more clear. Please see the Results section for details.

  1. Elaborate on how findings align or contrast with previous studies globally.

RESPONSE: It is now added at the end of the Discussion section. Text reads: “The present results are consistent with international evidence that underscores the important contribution of parental HL to managing chronic conditions, as well as en-hancing quality of life, of people with intellectual disabilities, including DS. Research from the United States and Western Europe has similarly found that higher caregiver HL status is correlated with better adherence to medical recommendations, better preventive care and fewer hospitalizations [65,66]. In line with trends in literature, our study found positive associations between higher parental HL and better metabolic and functional outcomes in adults DS [14]. However, in contrast to much of literature which has historically focused on children, our study imparts information on the role of HL in adults, which has been less represented in this context. While studies often look at HL in broader disability groups, this study is able to narrowly identify the DS population in a Croatian context, which adds new uncharted information on the DS population. The other finding of note was that there were no significant gender differences across var-ious domains, which is different from findings among larger international cohorts who found that females with DS sometimes have different health trajectories compared to males with DS [67]. Differences such as these may point to cultural, systemic or sample differences, which may warrant follow-up cross-cultural studies.”

  1. Discuss practical recommendations for improving parental HL further, including specific intervention strategies

RESPONSE: We thank the reviewer for this valuable suggestion. In response, we have expanded the Discussion and Conclusion sections to include practical, evidence-based recommendations for enhancing parental HL. These recommendations are based on existing international models and adapted to the Croatian healthcare and social context.

Text reads: “The results highlight how optimizing parental HL may facilitate support of the health of adults with DS. Possible tactics to improve parental HL in these settings may include conducting HL workshops for caregivers based on their various levels of literacy, routinely screening for HL in healthcare departments, and developing co-designed, user-friendly educational materials (e.g., visual guides, simplified instructions). Peer mentoring approaches that connect experienced caregivers with less experienced caregivers for informal learning can also help strengthen the informal learning networks. More importantly, engagement of interdisciplinary teams made up of healthcare providers, informal caregivers, educators, and policy makers can help institutionalize HL promotion as part of broader care systems. Investment in these types of practices can encourage families' capacity to manage medical complexity, improve continuity of care, and improve health outcomes for individuals with DS.”

Round 2

Reviewer 1 Report

Comments and Suggestions for Authors

Dear authors,

I stand by my comments on the interest and novelty of the research.
The ethical aspects have already been clarified.
However, more importantly, including all 25 DS in specific analyses — when only 18 had corresponding caregiver data — is methodologically problematic. If the 7 DS whose caregivers did not complete the assessments differ systematically from those included (e.g., if they were more or less obese), the assumption that the 25 DS represent a coherent sample is no longer valid. This introduces a clear risk of selection bias and may compromise the integrity of the analyses and conclusions drawn.

Therefore, only the 18 DS whose caregivers also provided complete data should be considered part of the analytical sample. Reporting results from all 25 children without this crucial context is misleading, especially when parent–child comparisons are central to the study.

As such, any reference to a sample size of 25 — in the abstract, methods, results, or elsewhere — should be revised to reflect the actual analytical sample of 18 matched pairs. 

Kind regards,

Author Response

Reviewer’s comments:
Dear authors,
I stand by my comments on the interest and novelty of the research.
The ethical aspects have already been clarified.
However, more importantly, including all 25 DS in specific analyses — when only 18 had
corresponding caregiver data — is methodologically problematic. If the 7 DS whose caregivers
did not complete the assessments differ systematically from those included (e.g., if they were
more or less obese), the assumption that the 25 DS represent a coherent sample is no longer
valid. This introduces a clear risk of selection bias and may compromise the integrity of the
analyses and conclusions drawn.
Therefore, only the 18 DS whose caregivers also provided complete data should be considered
part of the analytical sample. Reporting results from all 25 children without this crucial context
is misleading, especially when parent–child comparisons are central to the study.
As such, any reference to a sample size of 25 — in the abstract, methods, results, or elsewhere
— should be revised to reflect the actual analytical sample of 18 matched pairs.
In fact, here is a rewritten and condensed version in paragraph format:

RESPONSE: We thank the reviewer for her thoughtful and helpful critique. We agree
unequivocally that, when 18 of 25 participants with DS had accessible matched responses,
it was not methodologically transparent or good practice to include all 25 in analyses
requiring parental data, and this does raise legitimate concerns regarding selection bias.
In response, we agree to correct the manuscript so that only the 18 matched DS–parent
pairs are employed in any analyses. We made this explicit throughout the abstract,
methods, results, and discussion sections. Please note that the results of the health
parameters did not change much as the initial pairs who both fulfilled the questionnaire
and participated in the sampling remained the same. However, there was a difference in
the 6MWT results so we also included this in the discussion, text reads: “There were
notable sex differences in the assessment of exercise capacity (CRF) through the 6MWT,
with men scoring significantly higher than women. This is not a surprising result since

males often have a greater percentage of muscle mass and may have greater
cardiovascular capacity [38]. In DS adults, although both sexes have common limitations
of hypotonia, reduced aerobic capacity, and coordination deficits, males may still possess
physiological and functional benefits that promote walking performance [39]. Past studies
have also shown that DS males exhibit slightly higher physical activity and muscle
strength than females, which may influence endurance test results [9,40]. Behavioral
factors such as motivation, and self-confidence can also differ by sex and may contribute
to variability in performance [41]. These findings suggest that sex should be taken into
account when measuring cardiorespiratory fitness in adult persons with DS and are in
line with the value of individualized fitness testing and intervention.”

Thank you once again for providing valuable suggestions for improving our manuscript!